# Analysis of Excitation Energy Transfer in LaPO_4_ Nanophosphors Co-Doped with Eu^3+^/Nd^3+^ and Eu^3+^/Nd^3+^/Yb^3+^ Ions

**DOI:** 10.3390/ma16041588

**Published:** 2023-02-14

**Authors:** Karolina Sadowska, Tomasz Ragiń, Marcin Kochanowicz, Piotr Miluski, Jan Dorosz, Magdalena Leśniak, Dominik Dorosz, Marta Kuwik, Joanna Pisarska, Wojciech Pisarski, Katarzyna Rećko, Jacek Żmojda

**Affiliations:** 1Faculty of Electrical Engineering, Bialystok University of Technology, 45D Wiejska Street, 15-351 Bialystok, Poland; 2Faculty of Materials Science and Ceramics, AGH University of Science and Technology, 30 Mickiewicza Av., 30-059 Krakow, Poland; 3Institute of Chemistry, University of Silesia, 9 Szkolna Street, 40-007 Katowice, Poland; 4Faculty of Physics, University of Bialystok, K. Ciołkowskiego 1L, 15-245 Bialystok, Poland

**Keywords:** co-doped phosphors, I and II biological windows, optical properties, LaPO_4_, energy transfer, Eu^3+^, Nd^3+^, and Yb^3+^ ions

## Abstract

Nanophosphors are widely used, especially in biological applications in the first and second biological windows. Currently, nanophosphors doped with lanthanide ions (Ln^3+^) are attracting much attention. However, doping the matrix with lanthanide ions is associated with a narrow luminescence bandwidth. This paper describes the structural and luminescence properties of co-doped LaPO_4_ nanophosphors, fabricated by the co-precipitation method. X-ray structural analysis, scanning electron microscope measurements with EDS analysis, and luminescence measurements (excitation 395 nm) of LaPO_4_:Eu^3+^/Nd^3+^ and LaPO_4_:Eu^3+^/Nd^3+^/Yb^3+^ nanophosphors were made and energy transfer between rare-earth ions was investigated. Tests performed confirmed the crystal structure of the produced phosphors and deposition of rare-earth ions in the structure of LaPO_4_ nanocrystals. In the range of the first biological window (650–950 nm), strong luminescence bands at the wavelengths of 687 nm and 698 nm (^5^D_0_ → ^7^F_4_:Eu^3+^) and 867 nm, 873 nm, 889 nm, 896 nm, and 907 nm (^4^F_3/2_ → ^4^I_9/2_:Nd^3+^) were observed. At 980 nm, 991 nm, 1033 nm (^2^F_5/2_ → ^2^F_7/2_:Yb^3+^) and 1048 nm, 1060 nm, 1073 nm, and 1080 nm (^4^F_3/2_ → ^4^I_9/2_:Nd^3+^), strong bands of luminescence were visible in the 950 nm–1100 nm range, demonstrating that energy transfer took place.

## 1. Introduction

Nanophosphors, due to their unique structural and optical properties, are successfully used in biotechnology, biology, and medicine [1,2]. Among other things, they are applied as markers that have emission spectra in a specific range of wavelengths, in cancer therapy or diagnostics. Long radiative lifetimes of the excited levels in nanophosphors can be used to eliminate tissue autofluorescence. The structures of nanophosphors are distinguished by the close alignment of ions, which increases the probability of high energy transfer between them, thus limiting the concentration of quenching processes. With the use of nanostructures comes the possibility of, for example, obtaining thin, better-packed layers, which means a reduction in the amount of material needed [3,4,5,6,7,8,9,10]. Moreover, nanophosphors are characterized by the high sensitivity of detection and increased optical ranges of tissue penetration, which is an advantage in biomedical applications in the first (NIR—I: 650–950 nm) and the second (NIR—II: 1000–1350 nm) near-infrared biological windows. Compared to the visible range, NIR—I and NIR—II optical windows have a higher depth of tissue penetration and minimal scattering, as the degree of scattering in tissues decreases with increasing wavelength in almost all types of biological tissue [3,4,5,6,7,8,9,10]. The absorption of different wavelengths depends on the type of biological tissue. Water is almost transparent to visible light, but is characterized by strong absorption in the infrared region. Tissues such as blood and fat also have different absorption intensities in different wavelengths of light [3,11,12,13,14,15,16].

Over the last couple of years, special attention has been focused on photonic materials based on rare-earth (RE) ions. Due to the unique properties of Ln^3+^ such as long excited-state lifetimes, narrowing emission and absorption bands, large Stokes shifts, and rich 4f electronic energy level structures, they were widely used in many light-management applications. In addition, most lanthanide ions doped in inorganic host materials of REPO_4_ (RE = Y, La, Gd, Lu) have very high thermal and chemical stability [17,18,19,20,21,22] which is important in biosensing devices [23,24,25]. Among inorganic hosts, lanthanide phosphate (LaPO_4_) has a high index of refraction (*n* = 1.85) and is widely investigated as a biological-sensing, drug-delivery, and light-emitting device [26,27,28,29,30]. Lanthanum phosphate has also been identified as a suitable host matrix for lanthanide ions with the co-doping approach used to prepare compounds with tunable and multicolor emissions [31,32,33,34,35]. From a practical point of view, LaPO_4_-nanophosphor-doped europium ions have higher luminescence intensity in the range starting at 650 nm (*λ_exc_* = 395 nm) compared to Y_2_O_3_ [36]. In addition, the strong absorption of UV radiation by the LaPO_4_:Eu^3+^ leads to high quantum efficiency.

The most commonly described methods for producing LaPO_4_ doped with rare-earth ions include the following sol-gel, self-combustion, and hydrothermal methods, solid-state reactions, and wet chemical synthesis [19,37,38,39,40,41].

One of the rare earths with luminescence spectra in the first biological window is the europium ion. Due to the strong absorption of radiation in the 300–400 nm spectral range, triple-positive europium ions (Eu^3+^) are characterized by easy excitation from the ground state using relatively low optical power. Upon excitation of Eu^3+^ ions, the most intense emission bands are in the range from 580 to 720 nm, which is associated with the ^5^D_0_ → ^7^F_J_ transitions [42,43,44,45]. However, the width of emission bands of Eu^3+^ is limited from several to tens of nanometers. It is difficult to overlap the whole spectral range of biological windows by using a single RE dopant. Due to the presence of a large gap between the ^5^D_0_ level and the next-lower manifold (∼12,000 cm^−1^), there is the highest probability of energy transfer from the ^5^D_0_ level to the other level of the second element [46]. One of these elements can be neodymium or ytterbium ions which exhibit luminescence spectra in the first and second biological windows [47,48,49,50]. Neodymium ions (Nd^3+^) have emission bands at the wavelengths of 890 nm (^4^F_3/2_ → ^4^I_9/2_) and 1060 nm (^4^F_3/2_ → ^4^I_11/2_) [45,51]. Ytterbium ions (Yb^3+^) have a simple electronic level structure with one excited-state manifold ^2^F_5/2_ and ^2^F_7/2_ ground-state level (∼10,000 cm^−1^) with emission peaks at around 980 nm (^2^F_5/2_ → ^2^F_7/2_) [45,52,53].

The energy stored in the excited levels of the ions as a result of the absorption of pump radiation can be transferred to ions of the same type in the migration process or ions of a different type as a result of energy transfer. Non-radiative energy transfer depopulates the excited level of the excited ions, thus reducing the intensity of the luminescence. The use of two or more active dopants in some systems makes it possible to transfer some energy from the excited level to the lower state of the other elements in a radiative energy-transfer process. Ions that transfer the absorbed excitation energy are donors, while ions that absorb the energy of the excited donor are acceptors [54,55]. As a result of the analysis of the possibility of energy transfer between rare earths, it was established that energy transfer occurs between Eu^3+^ and Nd^3+^, where europium ions are donors and neodymium ions are acceptors. It is also possible to obtain energy transfer between Nd^3+^ and Yb^3+^ ions. Due to the energy transfers occurring between Eu^3+^ and Nd^3+^ and Nd^3+^ and Yb^3+^ there is the possibility of receiving luminescence bands in the VIS and NIR ranges [50,56,57,58].

In this work, we focus on the analysis of the excitation-energy transfer in LaPO_4_ nanophosphors fabricated by the co-precipitation method. Thus, the luminescence measurements of LaPO_4_:Eu^3+^, LaPO_4_:Eu^3+^/Nd^3+^, and LaPO_4_:Eu^3+^/Nd^3+^/Yb^3+^ samples in the range of the first and the second biological windows, were analyzed with excitation at 395 nm of laser radiation. In addition, we investigated radiative transitions in Eu^3+^, Nd^3+^, and Yb^3+^ and energy-transfer efficiency between RE ions. The efficiency of the energy transfer was determined by considering changes in luminescence intensity in the quantum structure of the donor as a function of interaction with the absorption field of the acceptor [59]. We also investigated the structural properties of co-doped nanophosphors.

## 2. Materials and Methods

The co-precipitation method, which does not require rigorous or rough synthetic conditions, was used to produce LaPO_4_ nanophosphors co-doped with RE ions. Initially, samples doped with one type of rare-earth ion were produced to select optimal concentrations for co-doped samples. The optimal concentrations 5 mol% of Eu_2_O_3_, 2 mol% of Nd_2_O_3_, and 5 mol% of Yb_2_O_3_ were chosen from our earlier work [43]. To synthesize the phosphors, La_2_O_3_, Eu_2_O_3_, Nd_2_O_3_, and Yb_2_O_3_ (Sigma-Aldrich, 99.99%), nitric acid (V), NH_4_H_2_PO_4_ (Chempur), and glycerol (Sigma-Aldrich, 99.5%) were used. To produce co-doped samples (LaPO_4_: Eu^3+^/Nd^3+^ and LaPO_4_: Eu^3+^/Nd^3+^/Yb^3+^), (93 − x) La_2_O_3_ − 5 Eu_2_O_3_ − 2 Nd_2_O_3_ − x Yb_2_O_3_, where x = 0 or 5 mol% were used in stoichiometric ratio. Firstly, rare earths were dissolved in nitric acid, and nitrates were mixed with 75 mL of deionized water and 25 mL of glycerin. Then the mixture was stirred until a temperature of 50 °C. The next step was to drip ammonium dihydrogen phosphate for 15 min and stir the resulting solution for 30 min at 50 °C. Next, the white suspension was centrifuged, and the precipitate was washed three times with deionized water and one time with ethanol. The sediments were placed on the filters and then put in the furnace at 80 °C for 42 h. In the last stage, the phosphors were annealed in the furnace at 1000 °C for 2 h and then ground.

Luminescence spectra were measured in the range of 650–950 nm for LaPO_4_: Eu^3+^/Nd^3+^ and 950–1100 nm for LaPO_4_:Eu^3+^/Nd^3+^/Yb^3+^ using an Acton 2300i monochromator (Acton Research Corporation, Acton, MA, USA) equipped with an InGaAs detector. We used laser radiation at the wavelength of 395 nm as the excitation source. A system PTI QuantaMaster QM40 (Horiba Instruments, New York, USA) coupled with tunable pulsed optical parametric oscillator (OPO), pumped by a third harmonic of a Nd:YAG laser (OpotekOpolette 355 LD, Carlsbad, USA) was used for luminescence-decay measurements. The laser system was equipped with a double 200 mm monochromator, and multimode UV-VIS PMT (R928) and Hamamatsu H10330B-75 detectors controlled by a computer. Luminescence-decay curves were recorded and stored by a PTI ASOC-10 (USB-2500) oscilloscope (Horiba, Northamption, UK) with an accuracy of ±1 µs.

The microstructure of the samples was investigated by analyzed by X-ray diffraction using PANalytical’s X’Pert Pro X-ray diffractometer (Almelo, The Netherlands). The radiation source was an X-ray tube with a linear focus and Cu anode. Diffraction patterns of samples were recorded in the range of 10 ÷ 90° 2θ angle at a rate of 0.05° 2θ/2 s.

The SEM-FEI Nova 200 NanoSEM scanning electron microscope (Hillsboro, OR, USA) with an EDS X-ray analyzer from EDAX (Pleasanton, CA, USA) was used to characterize the surface features and evaluate the morphological changes of samples. High vacuum conditions and an accelerating voltage of 18 kV were used. All samples were coated with a layer of carbon before measurement.

## 3. Results

### 3.1. Structural Characterization

The X-ray patterns of the prepared materials are shown in Figure 1. For the pure LaPO_4_, LaPO_4_:Eu^3+^, and co-doped materials with Eu^3+^/Nd^3+^ and Eu^3+^/Nd^3+^/Yb^3+^, it can be seen that the obtained samples were well crystallized. Some characteristic peaks centered at 2θ = 21.133°, 26.796°, 28.574°, 30.931°, 31.038°, 34.318°, 40.823°, 45.682°, 48.169°, 52.038°, and 52.752° were attributed to the (−1 1 1), (2 0 0), (1 2 0), (0 1 2), (−1 1 2), (−2 0 2), (0 3 1), (2 1 2), (1 0 3), and (1 3 2), and corresponded to the monoclinic monazite LaPO_4_ structure (P21/n as the space group, reference code: 01-083-0651). The presence of peaks in diffraction patterns confirm that LaPO_4_ was obtained in every sample. Photoluminescence measurements discussed later in the text revealed emissions that could be ascribed to rare-earth ions located in the LaPO_4_ [60]. The rare-earth doped monoclinic monazite phase of lanthanum phosphate (LaPO_4_) has been reported by Zhou [61], and Gavrilović [29].

The morphologies of the obtained LaPO_4_ nanophosphors co-doped with Eu^3+^/Nd^3+^ and Eu^3+^/Nd^3+^/Yb^3+^ ions were investigated by SEM (Figure 2a and Figure 3a). The SEM images of the samples shown in Figure 2a reveal that the particles were similar in shape. Moreover, from SEM images (Figure 3a), it can be seen that the powders were composed of short rods [62]. The diameter of the rods ranged from 180 nm to 690 nm.

Analysis of the chemical composition of produced nanophosphors by EDS (Figure 2b) with the weight percentage (Wt%) and atomic percentage (At%) of each element, confirms the presence of Eu, Nd, La, P, and O. The analysis presented in Figure 3b further shows the presence of Yb. The compositional information from the EDS suggests that rare-earth ions are embedded in the structure of LaPO_4_ nanocrystals.

Based on the research performed, it can be seen that the addition of ytterbium in the LaPO_4_:Eu^3+^/Nd^3+^/Yb^3+^ sample did not change the shape of the particles. It did, however, cause the particles to have smaller diameters up to 300 nm, compared to the LaPO_4_:Eu^3+^/Nd^3+^ sample.

### 3.2. Luminescence Properties

The luminescence spectra of fabricated LaPO_4_ nanophosphors doped with 5 mol% Eu^3+^ and co-doped with 5 mol% Eu^3+^ and 2 mol% Nd^3+^ are shown in Figure 4a. In the range of the first biological window (650–950 nm) two strong luminescence bands at the wavelengths of 687 nm and 698 nm originating from ^5^D_0_ → ^7^F_4_ transition were observed in the LaPO_4_:Eu^3+^ sample for an optimal doping concentration of 5 mol% Eu^3+^. In the case of LaPO_4_:Eu^3+^/Nd^3+^, additional emission sub-bands at the wavelengths of 867 nm, 873 nm, 889 nm, 896 nm, and 907 nm corresponding to ^4^F_3/2_ → ^4^I_9/2_ transition of Nd^3+^ ions were observed. In the co-doped sample, the intensity of the ^5^D_0_ → ^7^F_4_ transition rapidly decreased and these phenomena confirm the efficient energy transfer between Eu^3+^ and Nd^3+^ ions.

To illustrate the energy-transfer process between Eu^3+^ and Nd^3+^ ions the simplified energy level diagram with possible quantum transitions is shown in Figure 4b. After excitation at 395 nm, the europium ions were excited into the ^5^L_6_ energy state and then relaxed rapidly to the ^5^D_0_ level by multiphonon transitions. Europium ions had characteristic spectra at wavelengths of 687 nm and 698 nm, which were related to the radiative transition ^5^D_0_ → ^7^F_4_. Simultaneously, the energy could be transferred from the ^5^D_0_ level of europium ions to the ^4^G_5/2_ level of neodymium ions by non-radiative energy transfer, where multiphonon relaxations to the ^4^F_3/2_ state can occur. After that, emission spectra of neodymium in the first biological window were recorded for sub-bands at the wavelengths of 867 nm, 873 nm, 889 nm, 896 nm, and 907 nm corresponding to ^4^F_3/2_ → ^4^I_9/2_ radiative transition.

Figure 5a shows the luminescence spectra of LaPO_4_:Eu^3+^/Nd^3+^ and LaPO_4_:Eu^3+^/Nd^3+^/Yb^3+^ fabricated samples in the part range of the second biological window (950–1100 nm). Characteristic emission bands with Stark-splitting at the wavelengths of 1060 nm corresponding to the transition ^4^F_3/2_ → ^4^I_11/2_ (Nd^3+^) were observed in the LaPO_4_ co-doped with Eu^3+^/Nd^3+^ ions under 395 nm laser excitation. In comparison to the luminescence of LaPO_4_: Eu^3+^/Nd^3+^/Yb^3+^ sample additional emission bands at the wavelengths of 980 nm, 991 nm, and 1033 nm corresponding to ^2^F_5/2_ → ^2^F_7/2_ (Yb^3+^) transition were noticed. In addition, emission sub-bands at the wavelengths of 1048 nm, 1060 nm, 1073 nm, and 1080 nm related to the ^4^F_3/2_ → ^4^I_11/2_ radiative transition of neodymium ions, were observed.

In the case of the LaPO_4_ co-doped Eu^3+^/Nd^3+^/Yb^3+^ sample, energy transfer from Eu^3+^ to Nd^3+^, and from Nd^3+^ to Yb^3+^ occurred (Figure 5b). While exciting the produced co-doped nanophosphor with a UV laser diode (*λ* = 395 nm) luminescence spectra in the second biological window were analyzed. Excited europium ions can transit non-radiatively from ^5^L_6_ to lower energy levels until ^5^D_0_. Effective energy transfer can be obtained from the ^5^D_0_ level of europium to the ^4^G_5/2_ level to the transfer energy of neodymium. After non-radiative transitions to the ^4^F_3/2_ level, radiative transition at the wavelengths of 1048 nm, 1060 nm, 1073 nm, and 1080 nm was observed. Based on previous reports [57,63,64], the emission level for Nd^3+^ and the absorption level for Yb^3+^ are spatially overlapped and in the case of LaPO_4_:Eu^3+^/Nd^3+^/Yb^3+^ there is energy transfer between neodymium and ytterbium ions. Between the donor (Nd^3+^) and the acceptor (Yb^3+^) it is possible to obtain energy transfer from ^4^F_3/2_ level for Nd^3+^ ions to ^2^F_5/2_ level for Yb^3+^ ions. As a result, there was a radiative transition ^2^F_5/2_ → ^2^F_7/2_ in ytterbium and three separate emissions spectra at 980 nm, 991 nm, and 1033 nm were observed.

### 3.3. Efficiency of Energy Transfer

According to the excitation spectra (Figure 6) of fabricated nanophosphors, we determined the typical wavelengths that can be used to obtain efficient luminescence in the first and second biological windows. In the case of energy transfer from Eu^3+^ to Nd^3+^ ions, we chose the wavelength of 395 nm corresponding to the ^7^F_0_ → ^5^L_6_ (Eu^3+^) transition, and for energy transfer from Nd^3+^ to Yb^3+^ ions we selected the wavelength of 474 nm corresponding to the ^4^I_9/2_ → ^4^G_11/2_ (Nd^3+^) transition. We noticed that for co-doepd nanophosphor, when the monitoring wavelength is 1056 nm (Nd^3+^:^4^F_3/2_→^4^I_11/2_) the excitation spectra consisted of the transition for both RE ions. Thus, this effect can be explained as an effective energy transfer between europium and neodymium ions.

Figure 7a shows emission spectra of ^5^D_0_ → ^7^F_4_ transition for all fabricated samples under excitation at the wavelength of 395 nm. The apparent decrease in the intensity of the emission spectra in the LaPO_4_:Eu^3+^/Nd^3+^ sample compared to LaPO_4_:Eu^3+^ show the efficient energy transfer from europium to neodymium ions. Lower intensity is also seen in the LaPO_4_:Eu^3+^/Nd^3+^/Yb^3+^ sample compared to LaPO_4_:Eu^3+^/Nd^3+^, suggesting that an additional energy-transfer process from neodymium to ytterbium ions also occurred.

In order to determine the efficiency ($\eta_{ET}$) of energy transfer between europium and neodymium ions we analyzed the luminescence decays of Eu^3+^:^5^D_0_ level in all fabricated samples: LaPO_4_:Eu^3+^, LaPO_4_:Eu^3+^/Nd^3+^, and LaPO_4_:Eu^3+^/Nd^3+^/Yb^3+^ (Figure 7b) under excitation at the wavelength of 395 nm. In all cases, the decays were characterized by single-exponential behavior, which confirms direct Eu^3+^ → Nd^3+^ energy transfer. According to the results, the efficiency was estimated through the following equation:(1)$$\eta_{ET}=1-\frac{\tau_{EuNd}}{\tau_{Eu}}$$
where *η_ET_* is the energy-transfer efficiency and *τ_EuNd_* and *τ_Eu_* are the lifetimes of Eu^3+^ ions in LaPO_4_:Eu^3+^/Nd^3+^ and LaPO_4_:Eu^3+^, respectively. The calculated energy-transfer efficiency *η_ET_* was close to 60% for the LaPO_4_:Eu^3+^/Nd^3+^ sample and about 64% for LaPO_4_:Eu^3+^/Nd^3+^/Yb^3+^.

To better understand the possible mechanisms of energy transfer in the produced LaPO_4_:Eu^3+^, LaPO_4_:Eu^3+^/Nd^3+^, and LaPO_4_:Eu^3+^/Nd^3+^/Yb^3+^ samples, the critical distance (*R_c_*) was evaluated using Equation (2):(2)$$R_{c}=2{\left(\frac{3V}{4\pi X_{c}N}\right)}^{\frac{1}{3}}$$
where *R_c_* is the critical distance between the dopant ion and quenching site, *V* is the volume of the unit cell (for the LaPO_4_ host lattice, *V* = 304.86 Å^3^), *X_c_* is the critical concentration of RE ions *X*_c_ = 0.05, 0.07, and 0.12 for LaPO_4_:Eu^3+^, LaPO_4_:Eu^3+^/Nd^3+^, and LaPO_4_:Eu^3+^/Nd^3+^/Yb^3+^, respectively, and *N* is the number of cations per unit cell (*N* = 4). The critical distance was calculated as 14.28 for LaPO_4_:Eu^3+^, 12.77 for LaPO_4_:Eu^3+^/Nd^3+^, and 10.67 for LaPO_4_:Eu^3+^/Nd^3+^/Yb^3+^. In three cases the distances were larger than 10 Å which means the possible transfer was only between rare-earth ions. Additionally, three samples were characterized by single-exponential behaviour between Eu^3+^ ions, therefore there was a resonant energy transfer between Eu^3+^ and Nd^3+^. Optical parameters of LaPO_4_:Eu^3+^, LaPO_4_:Eu^3+^/Nd^3+^, and LaPO_4_:Eu^3+^/Nd^3+^/Yb^3+^ samples are summarized in Table 1.

In order to determine other possible energy transfers between Nd^3+^ and Yb^3+^ or Nd^3+^ and Eu^3+^ ions, we also analyzed the luminescence decay of ^4^F_3/2_ → ^4^I_9/2_ transition of Nd^3+^ ions measured for different excitation wavelengths. Firstly, as mentioned earlier, we selected the excitation wavelength of 474 nm and analyzed the luminescence spectra of LaPO_4_:Eu^3+^/Nd^3+^ and LaPO_4_:Eu^3+^/Nd^3+^/Yb^3+^ samples in the spectral range of 950–1100 nm (Figure 8a). The LaPO_4_:Eu^3+^/Nd^3+^/Yb^3+^ sample was characterized by a wide emission band originating from a superposition of radiative transitions of ytterbium ions and neodymium ions. Due to Nd^3+^ → Yb^3+^ energy transfer, the emission band at 1056 nm (Nd^3+^) had lower intensity than the LaPO_4_:Eu^3+^/Nd^3+^ sample.

Analogously, to determine the efficiency of Nd^3+^ → Yb^3+^ energy transfer we analyzed the lifetime of the ^4^F_3/2_ level of neodymium in LaPO_4_:Eu^3+^/Nd^3+^ and LaPO_4_:Eu^3+^/Nd^3+^/Yb^3+^. In this case, we also observed the single-exponent type of luminescence decay. Based on the above data (Figure 8b), the calculated energy-transfer efficiency *η_ET_* between neodymium and ytterbium was 46% for the LaPO_4_:Eu^3+^/Nd^3+^/Yb^3+^ sample. Moreover, based on another possible route of Nd^3+^ ion excitation, we measured luminescence decay from Nd^3+^: ^4^F_3/2_ of LaPO_4_:Eu^3+^/Nd^3+^ and LaPO_4_:Eu^3+^/Nd^3+^/Yb^3+^ samples with two different excitation wavelengths of 580 nm and 808 nm, and ^4^I_9/2_ → ^4^G_5/2_ and ^4^I_9/2_ → ^4^F_5/2_ transitions, respectively, were observed (Figure 9).

In both excitation schemes (580 nm and 808 nm) the Nd^3+^ → Yb^3+^ energy-transfer efficiency was close to 35%. In addition, the shapes of the decays were quite similar and they were characterized by single-exponential behaviour and it was confirmed that in fabricated nanophosphors the direct energy transfer from neodymium to ytterbium occurred. Table 2 summarizes lifetimes and calculated energy-transfer efficiency with different excitation of LaPO_4_:Eu^3+^/Nd^3+^ and LaPO_4_:Eu^3+^/Nd^3+^/Yb^3+^ samples.

The higher value of energy-transfer efficiency between neodymium and ytterbium at the excitation of 474 nm compared to the excitation of 580 nm and 808 nm suggests that the neodymium and europium ions were simultaneously excited. Based on the decay measurements (*λ_exc_* = 580 nm and *λ_exc_* = 808 nm), it can be concluded that only neodymium ions have effective energy transfer to ytterbium. Moreover, the similar values of energy transfer efficiencies (35%) at 580 nm and 808 nm excitation confirm that there are no other energy-transfer channels.

## 4. Conclusions

In this paper, detailed structure characterization and luminescence properties of LaPO_4_:Eu^3+^/Nd^3+^ and LaPO_4_:Eu^3+^/Nd^3+^/Yb^3+^ were performed. Using the co-precipitation method, it is possible to produce LaPO_4_ nanophosphors easily, without the need for stringent synthesis conditions. Co-doping the LaPO_4_ nanophosphors with rare-earth ions provides the possibility of achieving energy transfer between RE ions and obtaining the multiband emission in the first and second biological windows. Europium ions have strong absorption at the wavelength of 395 nm, which gives the possibility of using UV diodes to excite it efficiently. The use of europium ions as a donor shows the possibility of obtaining efficient energy transfer between europium and neodymium, and then between neodymium and ytterbium. The results of energy transfer were luminescence bands at the wavelength of 687 nm and 698 nm (^5^D_0_ → ^7^F_4_:Eu^3+^) and 867 nm, 873 nm, 889 nm, 896 nm, and 907 nm (^4^F_3/2_ → ^4^I_9/2_:Nd^3+^) in the 650–950 nm range, and 980 nm, 991 nm, 1033 nm (^2^F_5/2_ → ^2^F_7/2_:Yb^3+^) and 1048 nm, 1060 nm, 1073 nm, and 1080 nm (^4^F_3/2_ → ^4^I_9/2_:Nd^3+^) in the 950–1100 nm range. Based on the measurements performed, energy-transfer efficiency calculations were conducted. The calculated energy-transfer efficiency *η_ET_* from Eu^3+^ ions was close to 60% for the LaPO_4_:Eu^3+^/Nd^3+^ and about 64% for LaPO_4_:Eu^3+^/Nd^3+^/Yb^3+^. The energy transfer from Nd^3+^ to Yb^3+^ was about 46% with excitation at 474 nm and about 35% at 580 nm and 808 nm excitation. In addition, the critical distance (*R_c_*) was calculated and was 14.28, 12.77, and 10.67 for LaPO_4_:Eu^3+^, LaPO_4_:Eu^3+^/Nd^3+^, and LaPO_4_:Eu^3+^/Nd^3+^/Yb^3+^, respectively. Structural studies confirmed the occurrence of nanocrystals and the deposition of rare-earth ions in the structure of LaPO_4_ crystals.

Rare earths are characterized by narrow emission bands, making it necessary to look for additional/new opportunities that will give satisfactory results in terms of the first and second biological windows. To obtain broadband luminescence in the first and second biological windows, additional studies should be performed using rare earth ions having emission bands around 800 nm and/or 1100–1350 nm.

In studies conducted, the use of europium is shaping up as a universal approach, as luminescence in the range of the first and second biological windows can be achieved through co-doping and energy transfer. In addition, it is relatively simple to project the shape of the spectra with RE ions, which is a huge advantage.

## Figures and Tables

**Figure 1 materials-16-01588-f001:**
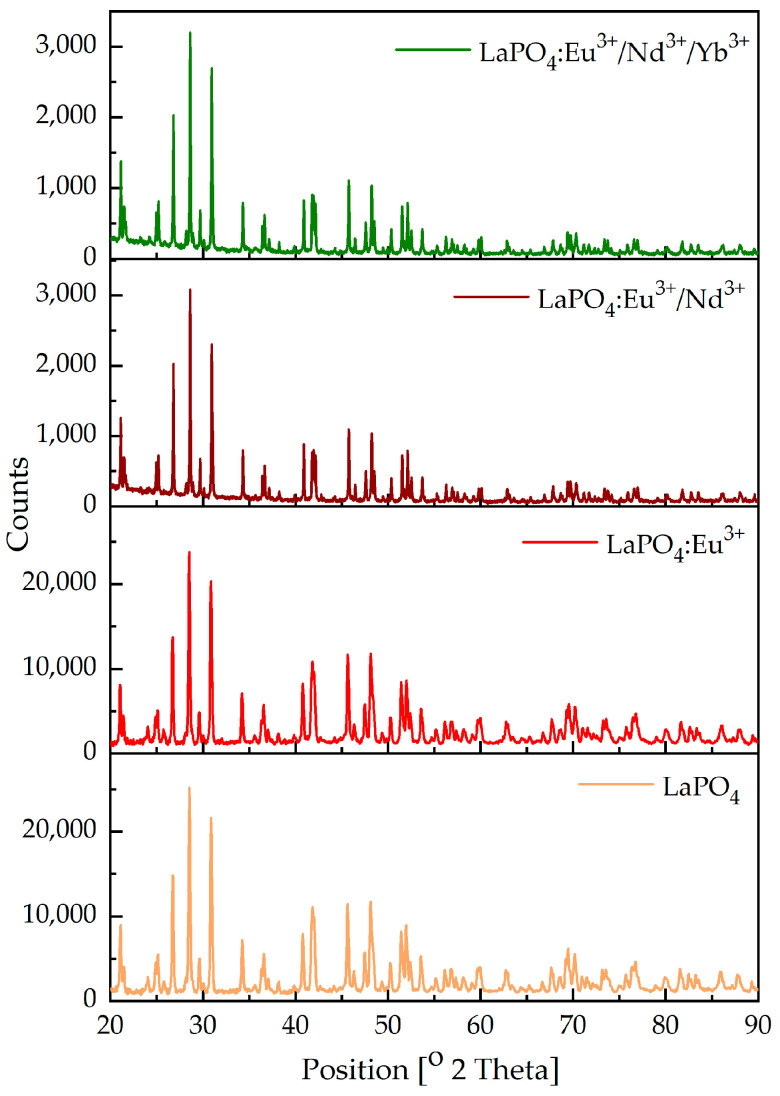
XRD measurements of LaPO_4_, LaPO_4_:Eu^3+^, LaPO_4_:Eu^3+^/Nd^3+^, and LaPO_4_:Eu^3+^/Nd^3+^/Yb^3+^.

**Figure 2 materials-16-01588-f002:**
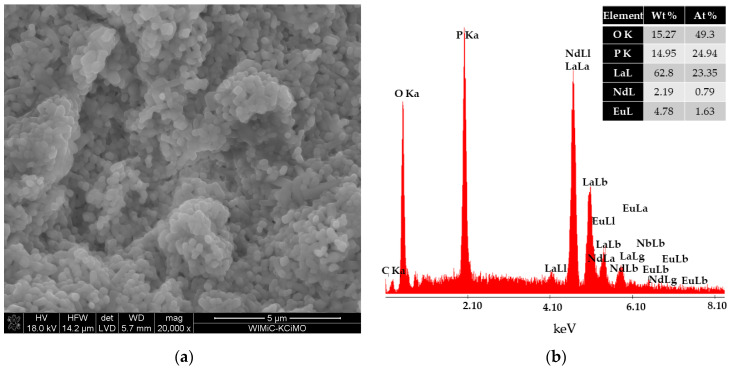
SEM image of LaPO_4_:Eu^3+^/Nd^3+^ with magnification 20,000× (**a**) and analysis of the chemical composition of the LaPO_4_:Eu^3+^/Nd^3+^ sample (**b**).

**Figure 3 materials-16-01588-f003:**
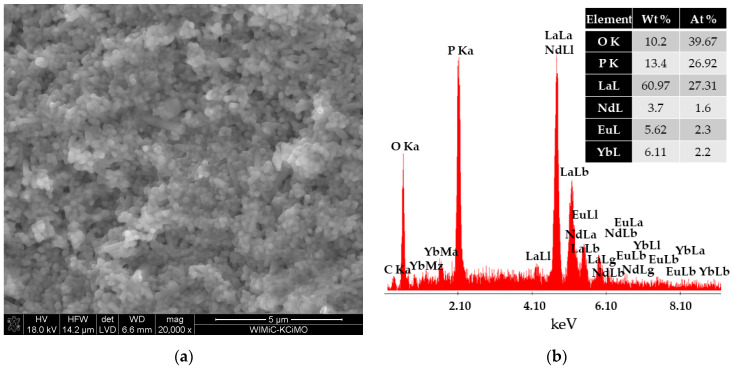
SEM image of LaPO_4_:Eu^3+^/Nd^3+^/Yb^3+^ with magnification 20,000× (**a**) and analysis of the chemical composition of the LaPO_4_:Eu^3+^/Nd^3+^/Yb^3+^ sample (**b**).

**Figure 4 materials-16-01588-f004:**
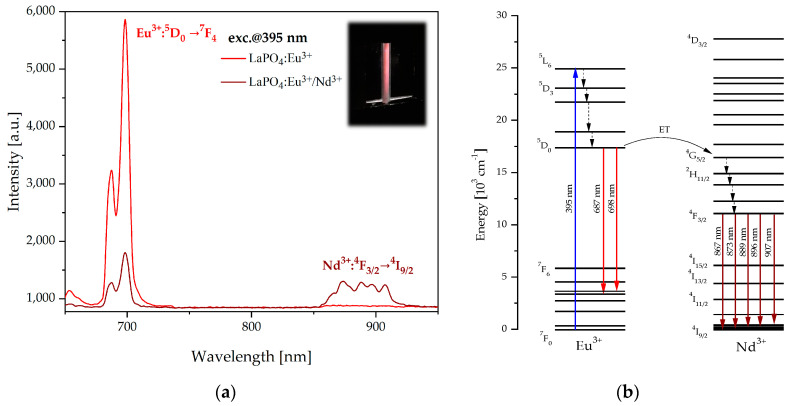
Luminescence spectra of LaPO_4_:Eu^3+^ and LaPO_4_:Eu^3+^/Nd^3+^ with inset of fluorescence sample LaPO_4_:Eu^3+^ (*λ_exc_* = 395 nm) (**a**) and simplified energy level diagram of Eu^3+^ and Nd^3+^ (**b**) illustrates the possible mechanism of energy transfer and radiative transitions in first biological window.

**Figure 5 materials-16-01588-f005:**
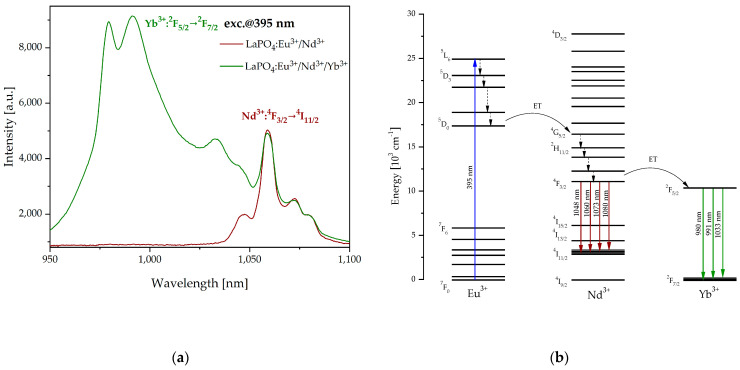
Luminescence spectra of LaPO_4_:Eu^3+^/Nd^3+^ and LaPO_4_:Eu^3+^/Nd^3+^/Yb^3+^ (**a**) and simplified energy level diagram of Eu^3+^, Nd^3+^, and Yb^3+^ (**b**) illustrates the possible mechanism of energy transfer and radiative transitions in second biological window.

**Figure 6 materials-16-01588-f006:**
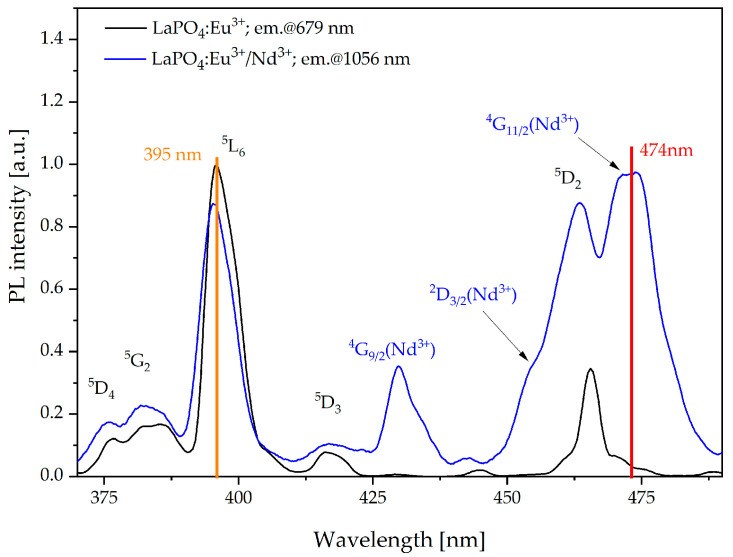
Photoluminescence (PL) spectra of LaPO_4_:Eu^3+^ (black line) and LaPO_4_:Eu^3+^/Nd^3+^ (blue line) nanophosphors.

**Figure 7 materials-16-01588-f007:**
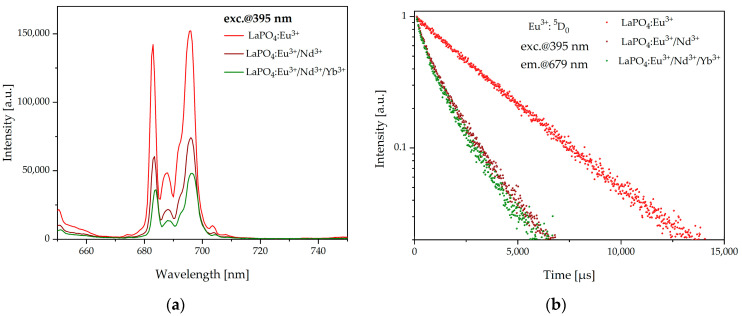
Luminescence spectra of LaPO_4_:Eu^3+^, LaPO_4_:Eu^3+^/Nd^3+^, and LaPO_4_:Eu^3+^/Nd^3+^/Yb^3+^ excited at 395 nm (**a**) and luminescence decay curves from Eu^3+^:^5^D_0_ (*λ_exc_* = 395 nm) of LaPO_4_:Eu^3+^, LaPO_4_:Eu^3+^/Nd^3+^, and LaPO_4_:Eu^3+^/Nd^3+^/Yb^3+^ samples (**b**).

**Figure 8 materials-16-01588-f008:**
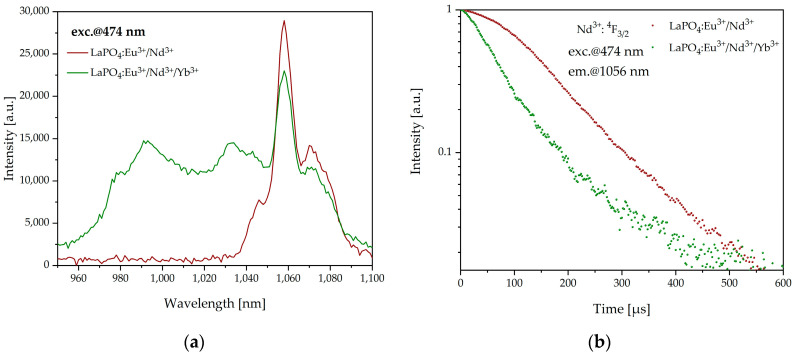
Luminescence spectra of LaPO_4_:Eu^3+^/Nd^3+^, and LaPO_4_:Eu^3+^/Nd^3+^/Yb^3+^ excited at 474 nm (**a**) and luminescence decay curves from Nd^3+^: ^4^F_3/2_ (*λ_exc_* = 474 nm) of LaPO_4_:Eu^3+^/Nd^3+^ and LaPO_4_:Eu^3+^/Nd^3+^/Yb^3+^ samples (**b**).

**Figure 9 materials-16-01588-f009:**
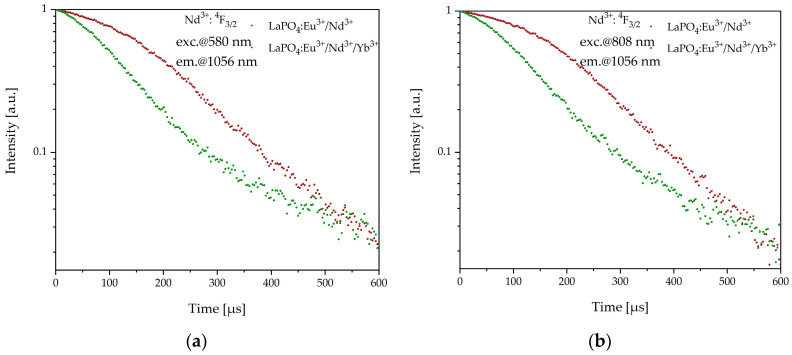
Luminescence decay curves from Nd^3+^: ^4^F_3/2_ (**a**) excited at 580 nm and (**b**) 808 nm of LaPO_4_:Eu^3+^/Nd^3+^ and LaPO_4_:Eu^3+^/Nd^3+^/Yb^3+^ samples.

**Table 1 materials-16-01588-t001:** Selected optical parameters of LaPO_4_:Eu^3+^, LaPO_4_:Eu^3+^/Nd^3+^, and LaPO_4_:Eu^3+^/Nd^3+^/Yb^3+^ samples.

Nanophosphor	Power (mW)	*λ_exc_* (nm)	Lifetime ^5^D_0_ (Eu^3+^) (ms)	$\eta_{ET\left(Eu\to Nd\right)}$ (%)	*R_c_* (Å)
LaPO_4_:Eu^3+^	10	395	3.247	---	14.28
LaPO_4_:Eu^3+^/Nd^3+^			1.293	60	12.77
LaPO_4_:Eu^3+^/Nd^3+^ /Yb^3+^			1.153	64	10.67

**Table 2 materials-16-01588-t002:** The spectroscopic parameters of LaPO_4_:Eu^3+^/Nd^3+^ and LaPO_4_:Eu^3+^/Nd^3+^/Yb^3+^ samples under different excitation wavelengths.

Parameters	LaPO_4_:Eu^3+^/Nd^3+^			LaPO_4_:Eu^3+^/Nd^3+^/Yb^3+^		
Excitation wavelength *λ_exc_* (nm)	474	580	808	474	580	808
Lifetime ^4^F_3/2_ (Nd^3+^) (ms)	0.141	0.186	0.194	0.076	0.120	0.128
$\eta_{ET\left(Nd\to Yb\right)}$ (%)	---	---	---	46	35	35

## Data Availability

Not applicable.

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
