# Peer review of "Analysis of Excitation Energy Transfer in LaPO4 Nanophosphors Co-Doped with Eu3+/Nd3+ and Eu3+/Nd3+/Yb3+ Ions"

_materials, 2023, doi:10.3390/ma16041588_

Round 1

Reviewer 1 Report

Comments:

Title: Analysis of excitation energy transfer in LaPO4 nanophosphors co-doped with Eu3+/Nd3+ and Eu3+/Nd3+/Yb3+ ions

Manuscript ID: materials-2130158

In the manuscript, authors have reported the analysis of the excitation energy transfer in LaPO4 nanophosphors fabricated by the co-precipitation method. The luminescence measurements of LaPO4:Eu3+, LaPO4:Eu3+/Nd3+, and LaPO4:Eu3+/Nd3+/Yb3+ samples in the range of the first and the second biological windows, were analyzed with excitation at 395 nm of laser radiation. However, after careful evaluation of this manuscript, I recommend it for publication in this journal after major revisions as suggested.

Authors can consider the following comments as given below,

 1.          The author must compare the XRD results of pure LaPO4 nanophosphors with LaPO4:Eu3+, LaPO4:Eu3+/Nd3+, and LaPO4:Eu3+/Nd3+/Yb3+ nanophosphors. Additionally, indexing the samples' XRD patterns.

2.          The author should investigate whether or not there was any lattice distortion because it appears in the host materials after doping.

3.          To understand how surface morphology changes with doping, authors should include TEM, and HRTEM, images of all the synthesized samples.

4.          To have a better understanding of the PL emission mechanism and charge transfer with doping, author should provide the photoluminescence decay profile of all the samples and calculate all the parameters (for example, critical distance etc.)

5.          The author must include digital images of the fluorescence sample while it is excited. Include the CIE chromaticity diagram.

Reviewer 2 Report

The manuscript describes the synthesis and characterization of doped nanophosphors based on LaPO4 matrix with emission in I and II biological windows of interest for diagnostic applications. Although the topic is very interesting and the current object of many studies, the work proposed here suffers from many weaknesses in terms of chemical and optical characterization. Furthermore, the novelty of this work with respect to what is described in the literature is not well documented and underlined by the authors. For these reasons, the manuscript is not suitable for publication at this stage.

Additional comments:

-  - The x-ray profiles supported by SEM micrographs reveal that the samples are composed of large particles, not really suitable for biomedical or diagnostic applications. Furthermore, there is no evidence on the particle size distribution of the particles in aqueous solution at physiological pH and on their colloidal stability. Both of these aspects are extremely relevant in view of further biomedical uses.

-    - The chemical composition of the particles is commented only from a qualitative point of view. It is very important to know the exact loading of Eu(III), Nd(III) and Yb(III) ions in the final samples. This information can be obtained by ICP or partially by EDS analysis. Without this information it is difficult to compare the optical properties and emission spectra of different samples. The intensity of the peaks also depends on the lanthanides amount confined in the inorganic matrix. To have more information on the energy transfer process, the analysis of the lifetimes of the excited state of the donor sites, alone, and in the presence of the acceptor species should be studied.

Reviewer 3 Report

The authors report on the chemical synthesis of co-doped rare earth nanoparticles based on Eu:Nd, and Eu:Nd:Yb for biological applications. The paper is well written and the experiment clear, however, I would advise  some minor revision before its publication in this journal.

- the authors should declare why they choose LaPO4 host compared to other, such as Y2O3 or other materials.

- Regarding the XRD spectrum, the reference spectrum from the database is not visible to me or not mentioned in the text.

- EDX spectra: a table should be provided reporting the ratio of the characteristic peaks associated with the elements or, even better, to evaluate the semiquantitative analysis.

- Please report a table befor the conclusion with the energy transfer efficiency, the optical excitation conditions, like power, lambda, .., and in particular the samples, as it not clear for the reader the differences among them in the paper.

Round 2

Reviewer 1 Report

The authors have addressed the comments. The manuscript can now be accepted for publication in this revised form.

Reviewer 2 Report

After careful review by the authors, the manuscript is now eligible for publication without further changes.